# SELECTIVE TOKEN GENERATION FOR FEW-SHOT LANGUAGE MODELING

## ABSTRACT

Natural language modeling with limited training data is challenging problem, and many algorithms make use of large-scale pretrained language models (PLMs) for this due to its great generalization ability. Among these transfer learning algorithms from PLMs, additive learning that incorporates a task-specific adapter on top of the fixed PLM has been popularly used to alleviate the severe overfitting problem in the few-shot setting. However, this added task-specific adapter is generally trained by maximum likelihood estimation that can easily suffer from the so-called exposure bias problem, especially in sequential text generation. Therefore, in this work, we develop a novel additive learning algorithm based on reinforcement learning (RL) for few-shot natural language generation (NLG) tasks. In particular, we propose to use a selective token generation between the transformer-based PLM and the task-specific adapter during both training and inference. This output token selection between the two generators allows the adapter to take into account only on the task-relevant parts in sequence generation, and therefore makes it more robust to overfitting as well as more stable in RL training. In addition, in order to obtain the complementary adapter from the PLM for each few-shot task, we exploit a separate selecting module that is also simultaneously trained using RL. Experimental results on various few-shot NLG tasks including data-to-text generation and text summarization demonstrate that the proposed selective token generation significantly outperforms the previous additive learning algorithms based on the PLMs.

## 1 INTRODUCTION

Natural language processing (NLP) have recently achieved great progress using advanced neural language models (Radford et al., 2018; Devlin et al., 2019; Yang et al., 2019; Liu et al., 2019; Clark et al., 2020; Raffel et al., 2019; Lan et al., 2019; Lewis et al., 2020). However, these neural models typically require large-scale training data for each individual task, and solving a new NLP task that has only a few examples is still challenging problem (Yin, 2020). Especially, natural language generation (NLG) with limited training data is an important yet more difficult task due to its fast adaptation of sequential prediction models in a wide range of applications including text summarization, question answering, data-to-text generation, machine translation, etc (Peng et al., 2020; Chen et al., 2020; Xu et al., 2021; Schick & Schütze, 2020; Chang et al., 2021; Radford et al., 2019; Lewis et al., 2020; Brown et al., 2020).

More recently, pretrained language models (PLMs) have shown great generalization ability when combined with large-scale data and big transformer-based models (Devlin et al., 2019; Radford et al., 2019; Lewis et al., 2020; Brown et al., 2020; Subramanyam Kalyan et al., 2021). Therefore, transfer learning from transformer PLMs has been popularly used for few-shot NLG tasks with promising results. In specific, the use of PLM for few-shot NLG can be categorized into three approaches: 1) prompt-based, 2) finetuning, and 3) additive learning. Prompt-based approaches encode a task description and task-specific examples as a natural language prompt for few-shot text generation (Radford et al., 2019; Brown et al., 2020; Zheng & Huang, 2021; Schick & Schütze, 2020; Li & Liang, 2021). While these approaches can take full advantage of the universal natural language understanding and generation capabilities of large-scale PLMs without further training of the main model, these have some limitations in dealing with a large domain shift from the pretraining corpus data, tuning suitable task-specific prompts, and covering an increased size of conditioning examples.

On the other hand, finetuning of the PLM is able to explicitly impart task-specific knowledge to the model and hence lift the above limitations (Ziegler et al., 2019; Xu et al., 2021; Chen et al., 2020). However, these finetuned models are prone to overfitting when only a small amount of training data is available. In order to alleviate such an overfitting problem, additive learning has been extensively exploited by incorporating task-specific adapters into the PLM (Zeldes et al., 2020; Stickland & Murray, 2019).

In general, task-specialized adapters for few-shot NLG are trained by maximum likelihood estimation (MLE). While MLE is efficient in learning, it suffers from the exposure bias problem due to the difference in the training and inference mechanisms (He et al., 2019), and this problem can be severe with limited training data. Reinforcement learning (RL) is capable of resolving this exposure bias problem by sequential output sampling during training (Ranzato et al., 2015; Keneshloo et al., 2019; Shi et al., 2021). Moreover, it allows to leverage the target-specific sequence-level objectives such as BLEU and ROUGE (Wu et al., 2018; Guo et al., 2021). However, in the task of NLG, the exponentially large space of output sequences restricts the use of RL since it leads to high variance and unstable training which is more serious in the few-shot setting. In this work, we develop a novel RL-based additive learning algorithm on the transformer-based PLM to overcome these shortcomings and to improve the performance of few-shot NLG.

In particular, we first convert the NLG task to the sequential token generation task based on the transformer language model, and then propose a selective token generation between the PLM and the task-specific adapter, during both RL-based training and inference. The proposed output token selection enables to not only explicitly maintain a general prior knowledge from the frozen PLM but also focus only on the task-relevant parts in sequence generation. In addition, in few-shot learning this partial token generation makes the task-specific adapter more resilient to overfitting and furthermore reduces the overall output space which leads to stable RL training. Here, in order to make the two token generators (policies) complement each other as well as to realize the robust output selection at the token level on the fly, we exploit a separate token-level policy selector. It is noted that both the policy selector and the task-specific adapter are simultaneously learned by the RL algorithm. Experimental results on various few-shot NLG tasks show that the proposed selective token generation outperforms the previous PLM-based additive learning algorithms with the comprehensive (non-selective) token generation.

Our main contributions can be summarized as follows.

- A novel selective token generation between the PLM and the task-specific adapter is proposed for transformer-based few-shot NLG.

- A separate selecting module is exploited to adaptively determine each output token in a sequence both at training and testing time.

- RL is applied to train both the policy selector and the task-specific adapter that is complementary to the PLM in text generation.

- An extensive empirical validation on few-shot NLG tasks demonstrates that the proposed selective token generation performs better in comparison to the previous PLM-based additive learning algorithms.

## 2 BACKGROUND

### 2.1 NATURAL LANGUAGE GENERATION

The goal of NLG is to generate a text sequence $\mathbf{y} = [y_0, ..., y_T]$ for a given task, where $y_t$ is the $t$th output token from a vocabulary $\mathcal{V}$, and $T$ is the output sequence length. For this generation, we aims to model the distribution of $\mathbf{y}$ that is autoregressively factorized as $p_\theta(\mathbf{y}) = \prod_{t=0}^{T} p_\theta(y_t|\mathbf{y}_{<t})$, where $\theta$ denotes the model parameters and $\mathbf{y}_{<t} = [y_0, ..., y_{t-1}]$. Here, the conditional distribution to sample a token for each step, $p_\theta(y_t|\mathbf{y}_{<t})$, is defined by the softmax function on the output logits $f_\theta(y_t|\mathbf{y}_{<t})$. Note that in general, the language generation is conditioned on input context according to a given task. Here, we encode the conditioning context by the same sequential model for generating an output sequence, and for simplicity we omit it. In this work, we utilize the autoregressive transformer for our generative model.

## 2.2 ADDITIVE LEARNING FOR FEW-SHOT GENERATION

To effectively leverage the general linguistic knowledge, $\theta$ is first initialized by the PLM parameters, $\theta_{LM}$, for NLG. Given $N$ task-specific training instances, $\mathcal{D} = \{\mathbf{y}^{n*}\}_{n=1}^{N}$, where $\mathbf{y}^{n*}$ is the $n$th ground-truth output sequence, directly finetuning $\theta_{LM}$ using $\mathcal{D}$ can incur the severe overfitting problem when $N$ is small in the few-shot scenario. Therefore, we add the task-specific adapter, $g_{\theta_a}$ parameterized by $\theta_a$, on top of the PLM, and optimize only $\theta_a$ (Zeldes et al., 2020; Stickland & Murray, 2019). In specific, we reformulate $f(\cdot|\mathbf{y}_{<t};\theta) = W^T h(\mathbf{y}_{<t};\theta_h)$ where $W \in \mathbb{R}^{H \times |\mathcal{V}|}$ and $h \in \mathbb{R}^H$ denote the weight matrix and the penultimate representations, respectively, and $\theta = \{W, \theta_h\}$. Then, we define the task-specific conditional distribution as follows:

$$p(y_t|\mathbf{y}_{<t};\theta_{LM},\theta_a) = softmax\left(W_{LM}{}^T h_{LM}(\mathbf{y}_{<t}) + W_a{}^T g\big(h_{LM}(\mathbf{y}_{<t});\theta_g\big)\right), \quad (1)$$

where $h_{LM}(\mathbf{y}_{<t}) = h(\mathbf{y}_{<t};\theta_{h,LM})$ and $\theta_a = \{W_a, \theta_g\}$. Here, the summation of the PLM logits and the adapter logits is motivated by auxiliary training[1]. It is noted that in our additive learning $\theta_a$ is updated while $\theta_{LM}$ is kept frozen. Hence, in the following we omit $\theta_{LM}$ such that $p_{\theta_a}(y_t|\mathbf{y}_{<t}) = p(y_t|\mathbf{y}_{<t};\theta_{LM},\theta_a)$ for simplicity.

## 2.3 MAXIMUM LIKELIHOOD ESTIMATION (MLE)

Given a small amount of training data $\mathcal{D} = \{\mathbf{y}^{n*}\}_{n=1}^{N}$, MLE optimizes $\theta$ by maximizing the data log-likelihood as follows:

$$\hat{\theta} = \arg\max_{\theta} \sum_{n=1}^{N} \sum_{t=0}^{T} \log p_\theta(y_t^{n*}|\mathbf{y}_{<t}^{n*}). \quad (2)$$

Here, the output token at each step is conditioned on not the previous sampled tokens from the current model but the previous ground-truth tokens $\mathbf{y}_{<t}^{n*}$. Namely, tokens are drawn from the data distribution during training, which is opposed to that tokens are drawn from the model distribution at test time. This discrepancy, also known as the exposure bias, leads that the errors will be accumulated along the generated sequence at test time since the model is biased to only perform well on the ground-truth history distribution. Especially, this bias problem can be more severe in the few-shot training. In addition, the token-level cross-entropy loss in MLE training is different from the sequence-level test metrics such as BLEU and ROUGE that are commonly used in the tasks of NLG.

## 2.4 REINFORCEMENT LEARNING (RL)

As an alternative to MLE, RL is able to overcome the exposure bias problem of MLE by sequence-level sampling from the model distribution during training (Ranzato et al., 2015). Also, RL can improve the performance by directly optimizing the evaluation metrics (Guo et al., 2021). In order to use RL for our additive learning, we reformulate our text generation as an RL problem: at each time step $t$, the agent takes the current state $\mathbf{s}_t = \mathbf{y}_{<t}$ as an input and performs an action $a_t$ that outputs a token $y_t$ by a policy $\pi_\theta(a_t|\mathbf{s}_t)$ corresponding to $p_\theta(y_t|\mathbf{y}_{<t})$. Then, the agent receives a reward $r_t = r(\mathbf{s}_t, a_t)$ and deterministically transitions to the next state $\mathbf{s}_{t+1}$. Here, note that the token-level intermediate reward $r_t = 0, \forall t < T$ when we use the delayed reward associated with the sequence-level evaluation metric between the two full sequences, $\mathbf{y}$ and $\mathbf{y}^*$. Let $\tau = \{(\mathbf{s}_t, a_t, r_t)\}_{t=0}^{T}$ be the trajectory generated by $\pi_\theta$. The RL objective for the optimal agent is to maximize the expected sum of future discounted rewards:

$$J(\pi_\theta) = \mathbb{E}_{\tau \sim \pi_\theta}\left[\sum_{t=0}^{T} \gamma^t r_t\right], \quad (3)$$

where $\gamma \in [0,1)$ is the discount factor.

---

[1]Although the auxiliary training is particularly designed for maximizing the likelihood of the target task output, it also can take an advantage for RL since the adapter logits are nearly zero before training is advanced. Namely, it lets the task-specific conditional distribution start learning from the distribution of PLM, not a uniform distribution.

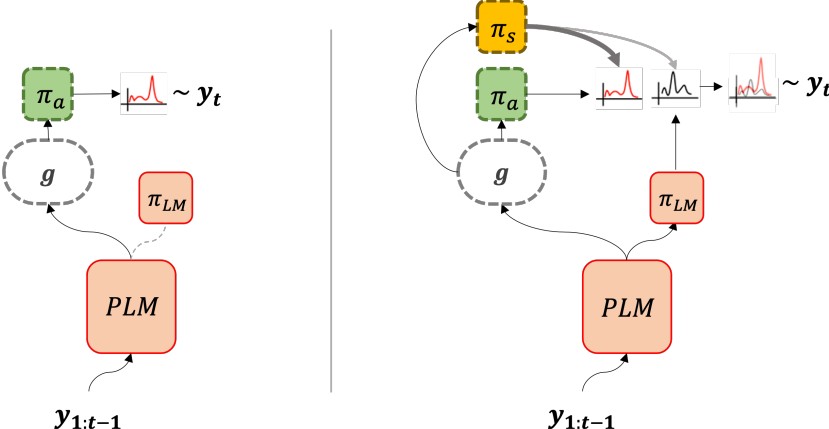

Figure 1: Text generation processes of Non-STG and STG are described. In the Non-STG model, every token is sampled from the task-specific policy $\pi_a$ (**Left**). On the other hand, in the proposed STG model, each token is selectively sampled from either the PLM policy $\pi_{LM}$ or the test-specific policy $\pi_a$ where the selection is performed by the selection policy $\pi_s$ (**Right**). Symbols with dashed line represent learnable models.

Among a number of algorithms to approximately optimize the RL objective, we employ an actor-critic algorithm (Bahdanau et al., 2017) since it explicitly optimizes the policy network and it can also alleviate the delayed reward problem. The actor-critic algorithm requires the additional critic network to estimate the value of a state, $V^\pi(\mathbf{s}_t) = \mathbb{E}_\pi[\sum_{t'=t}^T \gamma^{t'-t} r_{t'} | \mathbf{s}_t] = \sum_{a_t} \pi(a_t | \mathbf{s}_t) Q^\pi(\mathbf{s}_t, a_t)$ where the state-action value function $Q^\pi(\mathbf{s}_t, a_t) = \mathbb{E}_\pi[\sum_{t'=t}^T \gamma^{t'-t} r_{t'} | \mathbf{s}_t, a_t] = r_t + V^\pi(\mathbf{s}_{t+1})$. We use the following policy gradient loss to learn the policy parameters $\theta$:

$$\mathcal{L} = -\sum_{t=0}^T A^{\pi_\theta}(\mathbf{s}_t, a_t) \log \pi_\theta(a_t | \mathbf{s}_t), \tag{4}$$

where $A^{\pi_\theta}(\mathbf{s}_t, a_t) = Q^{\pi_\theta}(\mathbf{s}_t, a_t) - V^{\pi_\theta}(\mathbf{s}_t)$ is the advantage function that quantifies how an action $a_t$ is better than the average action in state $\mathbf{s}_t$.

In few-shot text generation, the extremely large action space ($|\mathcal{V}|^T$) as well as the small amount of training data often make it difficult to perform RL with degraded performances, even though we conduct the additive learning from the PLM. Furthermore, it commonly has a delayed reward function (e.g. BLEU) which is defined after an entire sequence generated. It is hard to decide which token and how much contributes to the reward. This problem is known as the credit assignment. Therefore, in this work, we propose a selective token generation for improving the RL-based additive learning.

## 3 SELECTIVE TOKEN GENERATION

Instead of generating all tokens in an output sequence from the single task-specific policy, $\pi_a = \pi_{\theta_a}(a_t | \mathbf{s}_t)$, at each time step $t$, we sample an output token $y_t$ selectively from either the PLM policy $\pi_{LM} = \pi_{\theta_{LM}}(a_t | \mathbf{s}_t)$ or the task-specific policy $\pi_a$:

$$y_t = a_t \sim \left( \mathbb{1}_t[\pi_{LM} \text{ is selected}] \pi_{LM}(a_t | \mathbf{s}_t) + (1 - \mathbb{1}_t[\pi_{LM} \text{ is selected}]) \pi_a(a_t | \mathbf{s}_t) \right), \tag{5}$$

where $\mathbb{1}_t[\cdot]$ is the indicator function (at $t$) that equals 1 if it is true and 0 otherwise. This output token selection allows to explicitly utilize a general linguistic knowledge from the PLM without catastrophic forgetting in few-shot learning. Also, the task-specific policy can focus on generating only the task-relevant parts, which enables more effective few-shot training with a reduced action space.

Now we need to determine how to select the proper policy at each step on the fly as well as to make the task-specific policy complementary to the PLM policy. For this, we exploit a separate

token-level policy selector. The proposed policy selector $\pi_s(i_t|\mathbf{s}_t; \theta_s)$ with the parameters $\theta_s$, where $i_t \in \{0, 1\}$, is an another policy that stochastically decides a policy to generate $a_t$ for $\mathbf{s}_t$. Namely, a token sample $y_t$ is generated by the following process:

$$i_t \quad \sim \quad \pi_s(i_t|\mathbf{s}_t), \tag{6}$$

$$y_t \quad = \quad \begin{cases} a_t \sim \pi_{LM}(a_t|\mathbf{s}_t) & \text{if } i_t = 0, \\ a_t \sim \pi_a(a_t|\mathbf{s}_t) & \text{if } i_t = 1. \end{cases} \tag{7}$$

This process can be considered as a token generation from a hierarchical policy $\pi_h(a_t|\mathbf{s}_t; \theta_s, \theta_{LM}, \theta_a)$ where the policy selector represents the upper-level prior for the preference of the low-level policy. Therefore, the value function of this hierarchical policy can be formulated as

$$V^{\pi_h}(\mathbf{s}_t) = \mathbb{E}_{\pi_h}[\sum_{t'=t}^{T} \gamma^{t'-t} r_{t'}|\mathbf{s}_t] \quad = \quad \pi_s(i_t = 0|\mathbf{s}_t) \sum_{a_t} \pi_{LM}(a_t|\mathbf{s}_t) Q^{\pi_h}(\mathbf{s}_t, a_t)$$

$$+ \pi_s(i_t = 1|\mathbf{s}_t) \sum_{a_t} \pi_a(a_t|\mathbf{s}_t) Q^{\pi_h}(\mathbf{s}_t, a_t), \tag{8}$$

and $A^{\pi_h}(\mathbf{s}_t, a_t) = Q^{\pi_h}(\mathbf{s}_t, a_t) - V^{\pi_h}(\mathbf{s}_t)$. Here, it is noted that a single critic network is used for the hierarchical policy since $i_t$ does not affect $\mathbf{s}_t$. Given a sample trajectory $\{(\mathbf{s}_t, i_t, a_t, r_t)\}_{t=0}^{T}$, the loss for optimizing $\theta_s$ and $\theta_a$ is

$$\mathcal{L} = -\sum_{t=0}^{T} A^{\pi_h}(\mathbf{s}_t, a_t) \bigg[ \mathbb{1}[i_t = 0]\big( \log sg[\pi_{LM}(a_t|\mathbf{s}_t)] + \log \pi_s(i_t|\mathbf{s}_t)\big)$$

$$+ \mathbb{1}[i_t = 1]\big( \log \pi_a(a_t|\mathbf{s}_t) + \log \pi_s(i_t|\mathbf{s}_t)\big) \bigg], \tag{9}$$

where sg stands for the stop-gradient operator. Similar to $\pi_a$, $\pi_s$ makes use of the PLM representations and the task-specific adapter such that

$$\pi_s(i_t|\mathbf{s}_t; \theta_s) = softmax\bigg( m\big(g(h_{LM}(\mathbf{s}_t)); \theta_s\big)\bigg), \tag{10}$$

where $m$ is the selector module that is implemented by a small neural network. Figure 1 depicts the overall text generation process by the proposed selective token generation (STG) in comparison to the previous non-selective token generation (Non-STG). Here, note that since all policies in STG share the same PLM representations, the increased computational cost by STG over Non-STG is negligible.

The use of the separated policy selector that is simultaneously trained with the task-specific policy allows the task-specific policy to be complementary to the PLM policy. Especially, this cooperative ensemble learning can be realized by our RL algorithm that performs sequential sampling from the model during training. In addition, the proposed STG model makes use of the PLM not at the feature level but the output distribution level in text generation. In our few-shot learning this is beneficial in explicitly retaining strong linguistic and world knowledge from the PLM.

## 4 EXPERIMENTS

In this section, we evaluate our method against baselines on Data-to-Text, Question Answering and Text Summarization tasks which are widely used in few-shot NLG. Since we claim the effectiveness of our method in additive learning, we compare our results against those obtained by other additive learning methods.

### 4.1 BASELINE

**PLM**. In our experiments, we assume that the PLM works to some extent for a given task. However, the naive PLM usually does not satisfy it for a new task unseen during training. Hence, we finetuned

Table 1: Data-to-Text performance on FewShotWOZ dataset.

| Model | Restaurant | | Hotel | | TV | | Laptop | |
|---|---|---|---|---|---|---|---|---|
| | BLEU $\uparrow$ | ERR $\downarrow$ | BLEU $\uparrow$ | ERR $\downarrow$ | BLEU $\uparrow$ | ERR $\downarrow$ | BLEU $\uparrow$ | ERR $\downarrow$ |
| PLM | 19.42 | 12.57 | 35.84 | 13.74 | 29.00 | 9.15 | 28.27 | 9.31 |
| Non-STG-MLE | 17.21 | 15.87 | 28.42 | 12.64 | 29.83 | 10.05 | 26.76 | 10.52 |
| Non-STG-RL | 18.01 | 11.98 | 36.72 | 12.64 | 28.66 | 9.19 | 28.59 | 9.21 |
| NE($max$)-MLE | 14.12 | 15.27 | 31.32 | 14.29 | 28.23 | 10.21 | 26.93 | 10.02 |
| NE($mix$)-MLE | **25.27** | 14.97 | 37.13 | 15.93 | **32.85** | 16.31 | **32.91** | 14.77 |
| NE($max$)-RL | 15.20 | 11.68 | 32.68 | 16.48 | 28.91 | 9.24 | 28.66 | 9.51 |
| NE($mix$)-RL | 24.10 | 19.16 | 38.07 | 18.68 | 32.84 | 18.06 | 32.53 | 17.14 |
| STG | 21.28 | **10.78** | **38.09** | **11.54** | 30.24 | **9.03** | 30.41 | **8.91** |

GPT-2[2] (Radford et al., 2019) with MLE for few epochs and used it as the PLM. Fine-tuning the PLM with MLE is most commonly used for task adaptation and thus it can also be a strong baseline. This fine-tuning phase accelerates the learning of the adapter. This is particularly when the adaptation requires to cover the large domain shift. Severe performance degradation was observed for all the tasks when we skipped the fine-tuning.

**Non-STG**. This method stands for Non-Selective Token Generation which uses the above the PLM as an encoder (frozen) and the adapter (additional layer to be trained). We use two objectives, MLE and RL, for additive learning. These will be denoted as Non-STG-MLE and Non-STG-RL, respectively.

**STG-Naive Ensemble**. We believe that our proposed method encourages the task-specific policy ($\pi_a$) to complement the PLM's policy ($\pi_{LM}$) with a proper selection of the selector through joint training. To investigate this, we evaluate against two different naive ensembles of the policies, $\pi_a$ trained from Non-STG and $\pi_{LM}$ of the PLM. These ensemble schemes are as follows:

- NE($max$): $\pi_{max} = softmax(Max(\pi_a, \pi_{LM}))$
- NE($mix$): $\pi_{mix} = (\pi_a + \pi_{LM})/2$

We also evaluate another naive ensemble strategy NE($random$) that randomly selects a token policy at each step between $\pi_a$ and $\pi_{LM}$, however it shows lower performances than the others.

## 4.2 IMPLEMENTATION

Here, we explain the implementation set-up of our method.

**Additional Layer**: The task-specific adapter $g$ in Section 2.2 is implemented by a LSTM to encode the dynamics of the representation vector $h_{LM}$. We found that the use of MLP was not performed well.

**Selector**. We use a 2-layer MLP with ReLU activation for $m$.

**Reinforcement Learning**. We employ Actor-Critic method (Konda & Tsitsiklis, 2000; Fedus et al., 2018) for RL. The agents (i.e. selector and generator) receive a reward after generating a sentence. Here, we use different reward functions according to tasks. We use delexicalised BLEU for Data-to-Text following Peng et al. (2020), Averaged score of BLEU and ROUGE-L for Question Answering and ROUGE-L for Text Summarization following Paulus et al. (2017) as the reward function.

**Token Sampling**. During the training, $i_t \in \{0, 1\} \sim \pi_s$ is first sampled, and then we use either $\pi_{LM}$ of the PLM for $i_t = 0$ or the task-specific policy $\pi_a$ for $i_t = 1$ to sample the $t$th token. During the evaluation, any decoding strategy, such as a beam search, can be used with the mixture of policies $\pi_h(\cdot) = \pi_s(i_t = 0)\pi_{LM}(\cdot) + \pi_s(i_t = 1)\pi_a(\cdot)$. We use the beam search decoding with a sample size of $k = 3$ for Text Summarization and $top_p = 0.9$ decoding for both Data-to-Text ($k = 10$) and Question Answering ($k = 3$).

---

[2]We make use of GPT-2 with 345M parameters as the initial checkpoint. We follow the training details in the previous works (Peng et al., 2020; Khandelwal et al., 2019) for each task.

Table 2: *Averaged performances* and *performance gains against the PLM* for Question Answering on $1\%$ few-shot subset data of MS-MARCO. The $gain$ indicates the averaged performance gain against the PLM's one with the *standard deviation* of the gain in the blacket. See the results of the other few-shots in Appendix A.

| Model | $BLEU$ | $BLEU_{gain}$ | $RL$ | $RL_{gain}$ | $Avg$ | $Avg_{gain}$ |
|---|---|---|---|---|---|---|
| PLM | 41.49 | 0.00($\pm$ 0.00) | 49.76 | 0.00($\pm$ 0.00) | 45.63 | 0.00($\pm$ 0.00) |
| Non-STG-MLE | 41.02 | -0.47($\pm$ 0.41) | 50.14 | 0.37($\pm$ 0.40) | 45.58 | -0.05($\pm$ 0.40) |
| Non-STG-RL | 41.25 | -0.24($\pm$ 0.34) | 49.97 | 0.20($\pm$ 0.21) | 45.61 | -0.02($\pm$ 0.27) |
| NE($max$)-MLE | 41.11 | -0.38($\pm$ 0.51) | 50.77 | 1.01($\pm$ 0.57) | 45.94 | 0.32($\pm$ 0.54) |
| NE($mix$)-MLE | 42.26 | 0.77($\pm$ 0.37) | **51.14** | **1.38**($\pm$ 0.40) | 46.70 | 1.08($\pm$ 0.38) |
| NE($max$)-RL | 41.51 | 0.02($\pm$ 0.47) | 50.54 | 0.78($\pm$ 0.38) | 46.03 | 0.40($\pm$ 0.41) |
| NE($mix$)-RL | 42.29 | 0.80($\pm$ 0.24) | 50.84 | 1.08($\pm$ 0.19) | 46.57 | 0.94($\pm$ 0.21) |
| STG | **42.76** | **1.27**($\pm$ 0.38) | **51.19** | **1.43**($\pm$ 0.37) | **46.98** | **1.35**($\pm$ 0.37) |

## 4.3 DATA-TO-TEXT

Data-to-Text is a task that transforms structured data such as graphs or tables into natural language. Recent works (Mager et al., 2020; Peng et al., 2020; Kale, 2020) show that the PLM can be adapted successfully to this task by taking a serialized form of data as an input without a carefully designed model to encode the structured data. Here, we perform experiments on FewShotWOZ (Peng et al., 2020) dataset. The evaluation is conducted only on the topics of Restaurant, Hotel, Laptop, and TV since the evaluator has not published yet for the other topics (Taxi, Attraction, Train)[3]. There are 50 training instances for each topic and 129, 78, 1379, and 680 testing instances for Restaurant, Hotel, Laptop, and TV, respectively. The models are evaluated by measuring fluency and informativeness using BLEU score and ERR (slot ERror Rate), respectively. Table 1 shows the obtained results.

## 4.4 LONG ANSWER QUESTION ANSWERING

We consider Long Answer QA task on MS-MARCO (Nguyen et al., 2016) dataset. In this task, a passage and a query are given, and the model generates an answer with respect to the query by referring to the passage. Here, we randomly sample various sizes of ($0.5\% \approx 500$, $1\%$, and $2\%$) subset data from the train dataset. We also sample a validation and a test set, which contains 500 and 12,000 instances, respectively, from the dev dataset. We repeat this test three times with different random seeds. In other words, we collect three different random subsets for each size of few-shot. Thus, we perform experiments on total nine subsets. The models are evaluated by measuring BLEU, ROUGE-L (denoted as $RL$), and their average value (denoted as $Avg$). The performances of the models are reported in Table 2 and Appendix A.

## 4.5 TEXT SUMMARIZATION

We consider the problem of abstractive summarization: models aim to generate its summary (the "target" text) for a given piece of "source" text such that its meaning is intact. The model has to learn 1) building a semantic form internally from the source text and 2) converting the semantic form into a text. Here, we randomly sample various sizes of ($0.5\% \approx 1,500$, $1\%$, and $2\%$) subset data from CNN/Daily Mail (See et al., 2017). We repeat this test three times for each size of few-shot as in above QA task. ROUGE (Lin, 2004) is commonly used to evaluate n-grams recall of the summaries with gold references. ROUGE-1, ROUGE-2, and ROUGE-L (measured based on the longest common sub-sequence) scores are reported (denoted as $R1$, $R2$, and $RL$, respectively) in Table 3 and Appendix B.

## 4.6 RESULT

We compare experimental results on various tasks in this subsection. In most cases, additive learning improves the performances over the PLM. However, they do not always guarantee a performance

---

[3]https://github.com/pengbaolin/SC-GPT

Table 3: *Averaged performances* and *performance gains against the PLM* for Text Summarization on $2\%$ few-shot subset data of CNN/DM. The $gain$ indicates the averaged performance gain against the PLM's one with the *standard deviation* of the gain in the blacket. See the results of the other few-shots in Appendix B.

| Model | $R1$ | $R1_{gain}$ | $R2$ | $R2_{gain}$ | $RL$ | $RL_{gain}$ |
|---|---|---|---|---|---|---|
| PLM | 33.05 | 0.00($\pm$ 0.00) | 12.96 | 0.00($\pm$ 0.00) | 23.39 | 0.00($\pm$ 0.00) |
| Non-STG-MLE | 33.19 | 0.14($\pm$ 0.08) | 12.98 | 0.02($\pm$ 0.05) | 23.39 | 0.00($\pm$ 0.06) |
| Non-STG-RL | 33.22 | 0.17($\pm$ 0.09) | 12.99 | 0.04($\pm$ 0.05) | 23.40 | 0.01($\pm$ 0.08) |
| NE($max$)-MLE | 33.19 | 0.14($\pm$ 0.10) | 12.99 | 0.03($\pm$ 0.05) | 23.40 | 0.01($\pm$ 0.06) |
| NE($mix$)-MLE | 33.11 | 0.06($\pm$ 0.04) | 12.99 | 0.03($\pm$ 0.03) | 23.41 | 0.02($\pm$ 0.03) |
| NE($max$)-RL | 33.21 | 0.16($\pm$ 0.10) | 12.99 | 0.04($\pm$ 0.06) | 23.41 | 0.02($\pm$ 0.08) |
| NE($mix$)-RL | 33.14 | 0.09($\pm$ 0.03) | 13.00 | 0.05($\pm$ 0.02) | 23.42 | 0.03($\pm$ 0.03) |
| STG | **33.45** | **0.40**($\pm$ 0.24) | **13.14** | **0.18**($\pm$ 0.10) | **23.66** | **0.27**($\pm$ 0.16) |

improvement. For example, the ERR score of the PLM on *Laptop* shows a better result except for STG and NE($mix$)-RL (see Table 1) and the Non-STG models trained on $1\%$ few-shot subset of MS-MARCO do not outperform the PLM (see Table 2 and Table 5 of Appendix A).

In Data-to-Text task, as shown in Table 1, we can observe that the Non-STG models do not outperform the PLM even though it has more neural units and takes more training time. The models trained on the RL objective show better performances for the ERR (lower is better). Interestingly, NE($mix$) methods show strong improvements for the BLEU which measures the fluency of sentence but obvious degeneration for the ERR which measures the rate of missing information from the given data. These results suggest that the PLM is much more capable of *task-general* knowledge than the *task-specific* generator (i.e. $\pi_a$) trained on few-shot dataset. It motivates us to jointly learn the policy selector and the task-specific generator. Here, while other methods show some trade-off between BLEU and ERR, only STG shows improvements on the both metrics for all topics in the dataset.

In Question Answering, NE($mix$) and STG show significantly better performances than the other methods as shown in Table 2 and Appendix A. Notably, NE($mix$) show good performances as much as STG. It obviously suggests that the PLM can be a complementary model to the additional model. Therefore, in this context, it can be lost of the prior knowledge of the PLM even the additional model has been built over the feature space of the PLM. We will discuss it in Section 4.7.

In Text Summarization (a problem of long-sequence generation), STG shows significantly larger gains than Non-STG-MLE, Non-STG-RL, and their naive ensembles with the PLM in every score metric and training data size as shown in Table 3 and Appendix B.

## 4.7 WHAT MAKES STG BETTER?

**Knowledge Preservation.** Although freezing the parameters of the PLM can mitigate the forgetting problem, it would be failed to do so when additional units having sufficient capacity is used since it is easy to memorize typical patterns of answers in small amount of training dataset and can degrade the generalization performance. An example that represent this issue is shown in Table 4 of Appendix A. In this case, a passage that contains two definitions (super-scripted and bolded) about *conflict* is given with a query which asks the psychological meaning of *conflict*. Without the knowledge of *who Colman*[4] *is*, it can be a hard to answer since the word *psychology* is not appeared in the passage. Here, the PLM repeats the given query due to the imperfect domain adaptation. Non-STG models, both Non-STG-MLE and RL, generate the same answer, that is not the psychological meaning but the general meaning of *conflict*, with pretty low perplexity ($\approx 0.35$). The reason in producing this incorrect answer is that most queries in few-shot training data ask a general meaning of a concept, and Non-STG models are overfitted to this pattern. On the other hand, the answer of STG, which is close to the ground truth, is generated by the PLM policy $\pi_{LM}$ after some sequence of tokens (*conflict is*) that are sampled from the task-specific policy $\pi_a$.

---

[4]A psychologist, https://en.wikipedia.org/wiki/Peter_T._Coleman_(academic)

We can find such examples for the other tasks in Appendix F: In Data-To-Text, as shown in the last example of Table 10, Non-STG generates *nicam stereo* which is not appeared in the given data. This is due to that *nicam stereo* was appeared 7 times (7/50, $14\%$) in training data. In Summarization, as shown in the first example of Table 14, Non-STG models only consider the forepart of the given article. Since the most of the major information is appeared in the forepart in News data, Non-STG models can be easily overfitted to generate the text according to such a pattern. Hence, we claim that Non-STG easily exposed learning patterns of typical answering and STG resolves this issue since it can be fully accessible to the knowledge of the PLM.

**Resolving Disadvantages of RL.** As described in Section 2.4, RL has some limitation with its application: 1) it suffers from exponentially large search space $|\mathcal{V}|^T$, and 2) it suffers from unstable training caused by credit assignment problem. STG resolves the first issue since the frozen PLM chooses a token when it is selected, and therefore the search space of the generator is approximately decreased from $|\mathcal{V}|^T$ to $|\mathcal{V}|^{T-\overline{T}_{PLM}}$ where $\overline{T}_{PLM}$ is the average length of sequences generated by PLM. For the second issue, the loss function of STG (Equation 9) intuitively shows that the gradient to the task-specific policy $\pi_a$ associated with producing $a_t$ will depend on the selector's action (i.e. $i_t = 1$). Hence, unlike Non-STG, $\pi_a$ of STG knows which token is contributed to the reward (see Appendix D for an illustration).

## 5   RELATED WORK

Recently, prompt-based in-context learning with an extremely large transformer-based PLM shows impressive few-shot generation performances (Radford et al., 2019; Brown et al., 2020). Schick & Schütze (2020) propose manually designed natural language prompts for improved few-shot text summarization and headline generation. Elsahar et al. (2018) conduct zero-shot learning for question generation from knowledge graphs, however they require a large amount of in-domain training data for their transfer learning. Chen et al. (2020) directly finetune the pretrained GPT-2 with a small amount of serialized attribute-value pairs for table-to-text generation. Gong et al. (2020) further apply multiple tasks to effectively leverage the structured information of tables. In contrast to these approaches, our proposed method utilizes RL-based additive learning for few-shot text generation.

Applying RL for text generation has been widely used to mitigate the exposure bias problem of MLE as well as to directly optimize task-relevant evaluation metrics. Ranzato et al. (2015) use the REINFORCE algorithm for text summarization and machine translation while Bahdanau et al. (2017) use the actor-critic algorithm for machine translation. However, they require pretraining using MLE. Ding & Soricut (2017) propose softmax policy gradient to remove the MLE-based pretraining. However, it requires various techniques for effective training. Tan et al. (2018) propose an entropy-regularized policy optimization that subsumes many of the previous training algorithms. Our proposed method is different from these methods in that we apply RL for more difficult few-shot generative modeling.

Various methods take into account the RL tasks with large action spaces like NLG. Dulac-Arnold et al. (2015) consider only actions in a cluster around the latent state of action obtained from a given state. Chandak et al. (2019) define the action embedding as a distribution with semantic of action and use a deterministic policy to take an action. Even-Dar et al. (2003); Zahavy et al. (2018) devise a method of incorporating the process of directly removing unnecessary actions according to the state in the RL problem. Guo et al. (2021) propose to use soft Q-learning and path consistency learning to combine off- and on-policy updates. Unlike these approaches, we use the hierarchical policy that reduces the sequential action space.

## 6   CONCLUSION

In this work, we propose to exploit a selective token generation between the pretrained language model and the task-specific adapter with RL-based additive learning for the tasks of few-shot natural language generation. In particular, we devise a trainable policy selector at the token level and jointly learn it with the task-specific policy. The proposed policy selector and RL algorithm make the two policies complementary each other and lead to robust few-shot generative modeling. Experimental results on various tasks of few-shot text generation show that the proposed selective token generation along with RL-based additive learning consistently improves the performances with less

overfitting. For future work, we will investigate more general ensemble learning for few-shot learning and perform more study on the architectures of both the adapter and selector.

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

# A    QUESTION ANSWERING RESULTS

Table 4: Generated answers from an instance of MS-MARCO QA dataset. Two definitions about *conflict* are presented in bold text in the passage. Tokens sampled from the task-specific policy $\pi_a$ are presented in red. The answers are sampled from the models trained on $0.5\%$ few-shot subset data.

| Passage | three types of conflicts are : 1. intrapersonal conflicts , 2. interpersonal conflicts and 3. unconscious conflicts . the word conflict has been derived from a latin word "conflicts" which means "strike two things at the same time" . conflict is [1)]**an opposition or a tug-of-war between contradictory impulses** . according to colman "a conflict is [2)]**the anticipated frustration entailed in the choice of either alternative**". |
|---|---|
| **Query** | conflict definition psychology |
| **Ground-truth** | the anticipated frustration entailed in the choice of either alternative. |
| **PLM** | conflict definition psychology. |
| **Non-STG** | conflict is an opposition or a tug-of-war between contradictory impulses. |
| **STG** | conflict is the anticipated frustration entailed in the choice of either alternative. |

Table 5: Averaged performances and performance gains against the PLM for Question Answering on $0.5\%$ few-shot subset data of MS-MARCO.

| Model | $BLEU$ | $BLEU_{gain}$ | $RL$ | $RL_{gain}$ | $Avg$ | $Avg_{gain}$ |
|---|---|---|---|---|---|---|
| PLM | 35.64 | 0.00($\pm$ 0.00) | 43.10 | 0.00($\pm$ 0.00) | 39.37 | 0.00($\pm$ 0.00) |
| Non-STG-MLE | 34.53 | -1.12($\pm$ 0.81) | 43.08 | -0.03($\pm$ 0.69) | 38.80 | -0.57($\pm$ 0.75) |
| Non-STG-RL | 35.08 | -0.56($\pm$ 0.24) | 42.78 | -0.32($\pm$ 0.14) | 38.93 | -0.44($\pm$ 0.17) |
| NE($max$)-MLE | 34.69 | -0.95($\pm$ 0.88) | 43.93 | 0.83($\pm$ 0.88) | 39.31 | -0.06($\pm$ 0.87) |
| NE($mix$)-MLE | 36.26 | 0.62($\pm$ 0.58) | 44.43 | 1.32($\pm$ 0.64) | 40.34 | 0.97($\pm$ 0.61) |
| NE($max$)-RL | 35.14 | -0.51($\pm$ 0.31) | 42.94 | -0.16($\pm$ 0.18) | 39.04 | -0.33($\pm$ 0.21) |
| NE($mix$)-RL | 35.93 | 0.29($\pm$ 0.20) | 43.52 | 0.42($\pm$ 0.29) | 39.73 | 0.35($\pm$ 0.24) |
| STG | **37.37** | **1.72**($\pm$ 0.34) | **44.53** | **1.43**($\pm$ 0.61) | **40.95** | **1.58**($\pm$ 0.40) |

Table 6: Averaged performances and performance gains against the PLM for Question Answering on $2\%$ few-shot subset data of MS-MARCO.

| Model | $BLEU$ | $BLEU_{gain}$ | $RL$ | $RL_{gain}$ | $Avg$ | $Avg_{gain}$ |
|---|---|---|---|---|---|---|
| PLM | 47.72 | 0.00($\pm$ 0.00) | 56.02 | 0.00($\pm$ 0.00) | 51.87 | 0.00($\pm$ 0.00) |
| Non-STG-MLE | 47.85 | 0.13($\pm$ 0.02) | 56.81 | 0.79($\pm$ 0.21) | 52.33 | 0.46($\pm$ 0.09) |
| Non-STG-RL | 48.00 | 0.28($\pm$ 0.33) | 56.83 | 0.82($\pm$ 0.31) | 52.42 | 0.55($\pm$ 0.32) |
| NE($max$)-MLE | 47.65 | -0.07($\pm$ 0.07) | 57.22 | 1.20($\pm$ 0.35) | 52.44 | 0.57($\pm$ 0.16) |
| NE($mix$)-MLE | **48.44** | **0.72**($\pm$ 0.22) | **57.30** | **1.28**($\pm$ 0.25) | **52.87** | **1.00**($\pm$ 0.23) |
| NE($max$)-RL | 47.58 | -0.14($\pm$ 0.57) | 57.06 | 1.05($\pm$ 0.57) | 52.32 | 0.45($\pm$ 0.56) |
| NE($mix$)-RL | 48.28 | 0.56($\pm$ 0.08) | 57.02 | 1.00($\pm$ 0.01) | 52.65 | 0.78($\pm$ 0.04) |
| STG | **48.42** | **0.70**($\pm$ 0.22) | **57.30** | **1.28**($\pm$ 0.38) | 52.86 | **0.99**($\pm$ 0.30) |

## B  SUMMARIZATION PERFORMANCE

Table 7: Averaged performances and performance gains against the PLM for Text Summarization on $0.5\%$ few-shot subset data of CNN/DM.

| Model | $R1$ | $R1_{gain}$ | $R2$ | $R2_{gain}$ | $RL$ | $RL_{gain}$ |
|---|---|---|---|---|---|---|
| PLM | 30.19 | 0.00($\pm$ 0.00) | 11.27 | 0.00($\pm$ 0.00) | 21.21 | 0.00($\pm$ 0.00) |
| Non-STG-MLE | 30.34 | 0.14($\pm$ 0.13) | 11.32 | 0.05($\pm$ 0.07) | 21.20 | -0.01($\pm$ 0.06) |
| Non-STG-RL | 30.35 | 0.15($\pm$ 0.15) | 11.34 | 0.07($\pm$ 0.07) | 21.22 | 0.02($\pm$ 0.08) |
| NE($max$)-MLE | 30.33 | 0.14($\pm$ 0.11) | 11.31 | 0.04($\pm$ 0.06) | 21.20 | 0.00($\pm$ 0.04) |
| NE($mix$)-MLE | 30.32 | 0.13($\pm$ 0.08) | 11.31 | 0.04($\pm$ 0.04) | 21.23 | 0.02($\pm$ 0.04) |
| NE($max$)-RL | 30.37 | 0.18($\pm$ 0.11) | 11.35 | 0.08($\pm$ 0.05) | 21.26 | 0.06($\pm$ 0.02) |
| NE($mix$)-RL | 30.28 | 0.09($\pm$ 0.09) | 11.30 | 0.03($\pm$ 0.05) | 21.22 | 0.02($\pm$ 0.04) |
| STG | **30.47** | **0.28**($\pm$ 0.15) | 11.37 | 0.09($\pm$ 0.05) | **21.36** | **0.15**($\pm$ 0.09) |

Table 8: Averaged performances and performance gains against the PLM for Text Summarization on $1\%$ few-shot subset data of CNN/DM.

| Model | $R1$ | $R1_{gain}$ | $R2$ | $R2_{gain}$ | $RL$ | $RL_{gain}$ |
|---|---|---|---|---|---|---|
| PLM | 31.03 | 0.00($\pm$ 0.00) | 11.56 | 0.00($\pm$ 0.00) | 21.62 | 0.00($\pm$ 0.00) |
| Non-STG-MLE | 31.25 | 0.22($\pm$ 0.03) | 11.63 | 0.07($\pm$ 0.02) | 21.72 | 0.10($\pm$ 0.07) |
| Non-STG-RL | 31.24 | 0.21($\pm$ 0.13) | 11.64 | 0.08($\pm$ 0.05) | 21.76 | 0.13($\pm$ 0.12) |
| NE($max$)-MLE | 31.25 | 0.22($\pm$ 0.04) | 11.63 | 0.07($\pm$ 0.03) | 21.72 | 0.09($\pm$ 0.08) |
| NE($mix$)-MLE | 31.20 | 0.17($\pm$ 0.04) | 11.61 | 0.04($\pm$ 0.02) | 21.69 | 0.06($\pm$ 0.04) |
| NE($max$)-RL | 31.26 | 0.24($\pm$ 0.13) | 11.66 | 0.10($\pm$ 0.04) | 21.78 | 0.15($\pm$ 0.12) |
| NE($mix$)-RL | 31.18 | 0.15($\pm$ 0.08) | 11.63 | 0.06($\pm$ 0.02) | 21.69 | 0.07($\pm$ 0.06) |
| STG | **32.03** | **1.00**($\pm$ 0.48) | **11.96** | **0.40**($\pm$ 0.15) | **22.23** | **0.61**($\pm$ 0.33) |

## C  TRAINING SETTINGS

In our experiments all the models of additive learning, Non-STG and STG, are used the same architecture and hyper-parameters (except whether to use pre-training) for training as described in Table 9. We found that pre-training the addtional layer of Non-STG-RL with MLE helps the performance improvements. On the other hand, STG without pre-training shows better performances. We use the training data for each topic of the task of Data-to-Text as their validation data.

Table 9: **Hyper-parameters used for experiments**

| Hyper-parameter | Summarization | Data-to-Text | Question Answering |
|---|---|---|---|
| Num layer | | 2 | |
| RNN hidden size | 512 | 256 | 256 |
| $\gamma$ | | 1 | |
| Optimizer | AdamW with $betas = (0.9, 0.999)$, $eps = 10^{-8}$ | | |
| Learning rate | 2e-5 | | 5e-5 |
| Pre-train epoch (Non-STG-RL) | 1 | 0 | 1 |
| Validation | 500 | 50 | 500 |
| Train epochs | 16 (0.5%), 8 (1%), 4 (2%) | 30 | 20 (0.5%), 10 (1%), 5 (2%) |
| Batch size | 16 | 10 | 16 |

## D    ILLUSTRATION OF STG

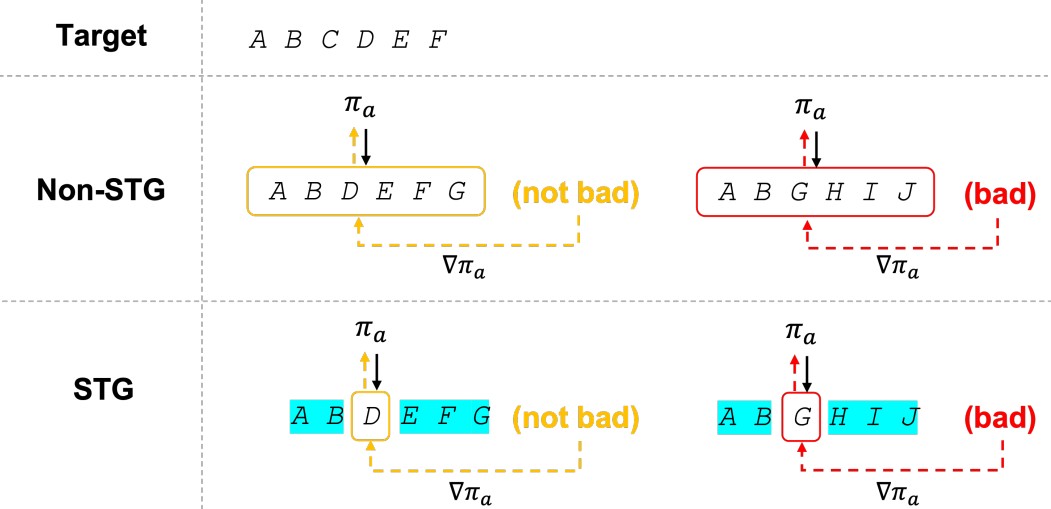

Figure 2: A simple schematic illustration of Non-STG and STG. **Non-STG(RL):** the whole sequence of target is generated from the task-specific policy $\pi_a$ so the right sub-sequence AB is also penalized from the delayed feedback. **STG:** the third token is sampled from $\pi_a$ and the model lets the other tokens (highlighted with cyan) generated from the PLM's policy $\pi_{LM}$ which generates a next letter of the previous alphabet input. Here, $\pi_a$ will be penalized at the third token.

## E    ADDITONAL STUDY

**STG-MLE.** Here, we evaluate the MLE version of STG (denoted as STG-MLE) which is trained by MLE for the mixture policy $\pi_h(\cdot) = \pi_s(i_t = 0)\pi_{LM}(\cdot) + \pi_s(i_t = 1)\pi_a(\cdot)$ similar to *copy mechanism* (Gu et al., 2016). In few-shot training, the explicit use of PLM logits can efficiently reduce the fine-tuning loss especially when the adapter is light since the adapter can focus only on the task-relevant part in generation. STG-RL[5] learns to do this naturally by stochastic policy sampling if the policy selector is initialized to perform uniform sampling. On the other hand, STG-MLE can be easily collapsed to select only a task-specific policy (i.e. $i_t = 1$). This is because the gradient flows the additional model only and, unlike STG-RL, there is no chance to exploit diverse paths during training in the teacher forcing manner. As shown in Figure 4, STG-MLE starts from the same point of STG-RL but it collapsed to Non-STG-MLE.

**Learning Curve.** It is well known that the RL-tuning resolves the exposure bias of MLE-tuning. We can expect that an additive learner of MLE would be affected by the exposure bias as well, and the RL objective for additive learning resolves it. Here, we present some learning curves[6] obtained from training in our experiments. As shown in Figure 4, the learners of MLE seem to have overfitting (in terms of Perplexity, PPL) and exposure bias (in terms of Score). On the other hand, the learners of RL were less effected by the problems. We can find that the STG models (denoted STG-RL) are superior to the others from the perspective of the score.

**Effectiveness of Selector.** Here, we investigate the effectiveness of the selector $\pi_s$ of the STG. We compare *Fixed Selection* against the *Dynamic selection*. In the fixed selection, the probability of selecting the PLM's policy $\pi_{LM}$ is fixed to $\pi_s(0_t|s_t) = 1 - \pi_s(1_t|s_t)$. We measure the performance with respect to $\pi_s(1_t|s_t) = c$ where $c$ is a constant. The selection will be uniformly random when $c = 0.5$, and when $c = 0$, the performance will be equivalent to the performance of the PLM without additive learning. Figure 3 shows that the input-dependent dynamic selection by our STG

---

[5]We add "-RL" to the STG to distinguish with STG-MLE in this context.
[6]The curve for Data-to-Text is not presented since there is no actual validation set.

outperforms the fixed selection with any $c$. We can find that how $\pi_s$ works for each task. For instance, in QA task, the first few tokens of an answer may decide the quality of generation (i.e. "yes" or "no" in binary QA). Therefore, an optimal strategy of the STG might be producing the first few tokens sampled from the task-specific $\pi_a$ and the remaining tokens from the PLM $\pi_{LM}$. The curve supports this interpretation since the score is decreased as $c$ is close to 1. Our STG learns such a strategy as shown from the generated answers in Table 11 and 12. In Data-to-Text, the BLEU score is increased as $c$ is close to 1 while the ERR score is decreased. This fact supports the results of NE($mix$) models as discussed in Section 4.6. The $\pi_s$ learns to balance between the BLEU and ERR.

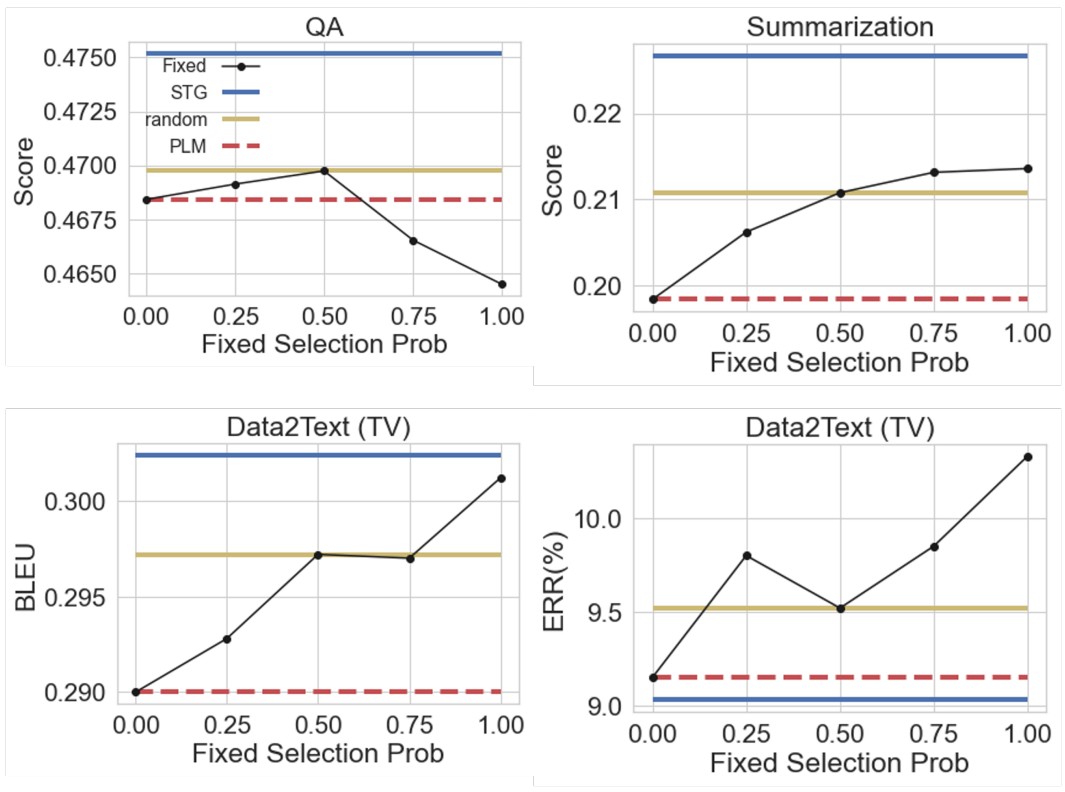

Figure 3: Dynamic selection vs Fixed selection.

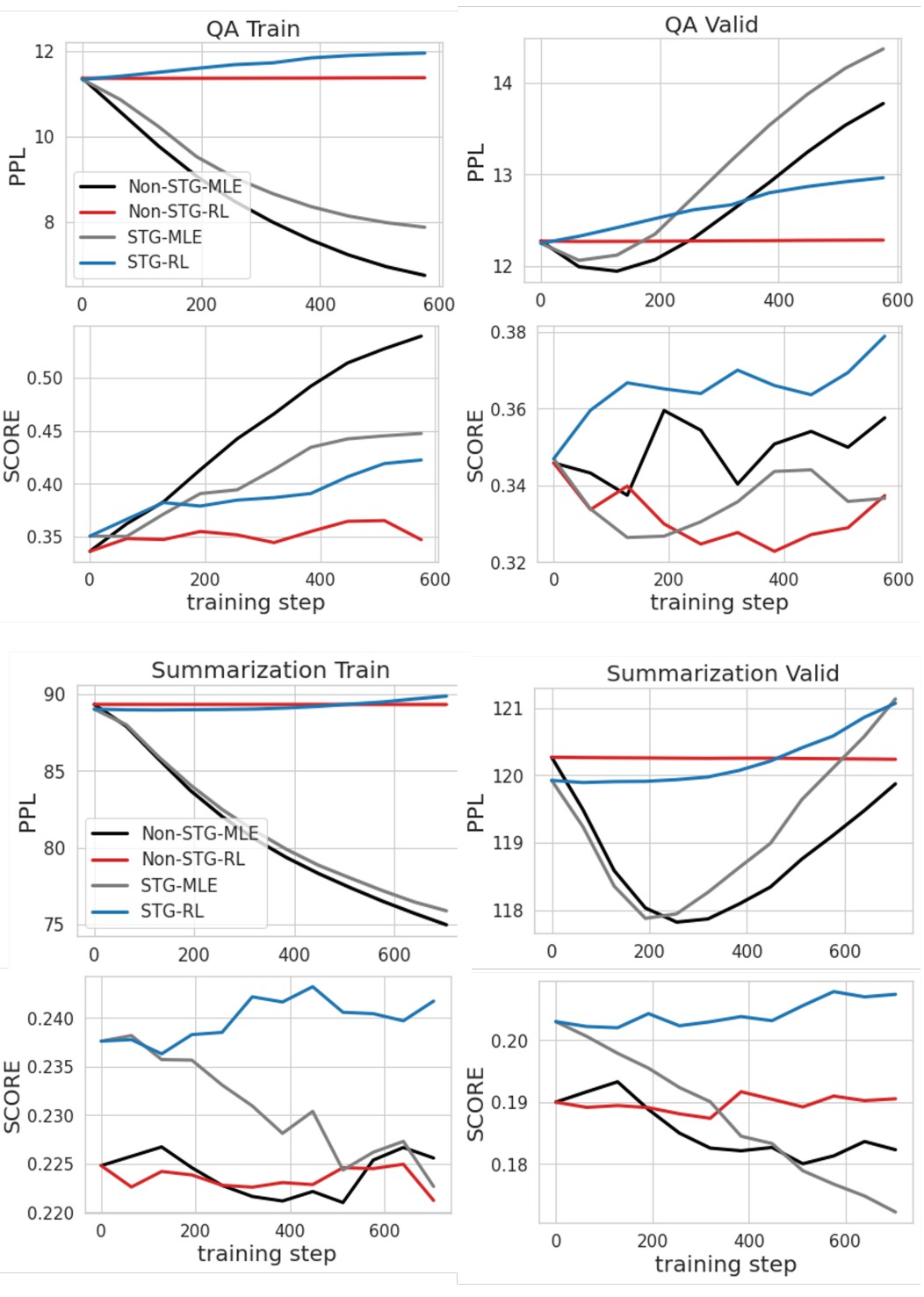

Figure 4: The learning curve. The Perplexity (PPL) and Score of each task (e.g. Rouge-L for Summarization) are measured on the 0.5% few-shot train set and the valid set.

# F  GENERATED SENTENCE EXAMPLES

Here, we show generated sentence examples for each task (see Table 10 for Data-to-Text, Table 11 and Table 12 for Question Answering and Table 13 and Table 14 for Summarization.). The tokens sampled from the task-specific policy $\pi_a$ are presented in red.

Table 10: Data-to-Text examples from TV of FewShotWOZ dataset.

| | |
|---|---|
| **Data** | recommend(name=kratos 31; type=television; powerconsumption=31.4 watt; accessories=remote control and european warranty; color=mat black bezel with mat black frame stand) |
| **Ground-truth** | the kratos 31 is a great television . it only consumes 31.4 watts , comes with a remote control and european warranty and has a stylish mat black bezel with mat black frame stand . |
| **PLM** | the kratos 31 features a mat black bezel with mat black frame stand and european warranty. |
| **Non-STG-MLE** | the kratos 31 features a mat black bezel with mat black frame stand and european warranty |
| **NE($mix$)-MLE** | the kratos 31 television has 31.4 watt power consumption, remote control and european warranty. |
| **STG** | the kratos 31 television has 31 watt power consumption. it comes with a remote control and european warranty with a color of mat black bezel with mat black frame stand. |
| **Data** | inform(name=triton 52; type=television; family=l7; audio=cevo stereo; screensize=47 inch) |
| **Ground-truth** | the triton 52 47 inch screen television from the l7 product family and has cevo stereo audio. |
| **PLM** | the triton 52 television is a 47 inch television in the l7 family and with a cevo stereo. |
| **Non-STG-MLE** | the triton 52 television is in the l7 family, it has 47 inches of screen, and uses cevo stereo sound capabilities. |
| **NE($mix$)-MLE** | the triton 52 television is in the l7 family, it has 47 inches of screen, and a 47 euro price, |
| **STG** | the triton 52 television is a 47 inch television in the l7 family and has a cevo stereo. |
| **Data** | inform(name=attis 65; type=television; resolution=1080p; pricerange=cheap) |
| **Ground-truth** | the attis 65 television is in the cheap price range and has a nice resolution of 1080p. |
| **PLM** | the attis 65 television in the cheap price range has a 1080p resolution and a pricerange of cheap. |
| **Non-STG-MLE** | the attis 65 television has a 1080p resolution, is in the cheap price range and has a nicam stereo. |
| **NE($mix$)-MLE** | the attis 65 television is a 1080p television in the cheap price range and in the cheap price range with a nicam stereo. |
| **STG** | the attis 65 television is in the cheap price range and has a 1080p resolution. |

Table 11: Question Answering examples.

| Passage | zygote : this cell results from the union of an oocyte and a sperm . a zygote is the beginning of a new human being ( i.e. , an embryo ) . the expression fertilized ovum refers to a secondary oocyte that is impregnated by a sperm ; when fertilization is complete , the oocyte becomes a zygote.10 ( emphasis added . ) |
|---|---|
| Query | what is a human zygote |
| Ground-truth | the beginning of a new human being . |
| PLM | the expression fertilized ovum refers to a secondary oocyte that is impregnated by a sperm ; |
| NE($mix$)-MLE | the expression fertilized ovum refers to a secondary oocyte that is impregnated by a sperm ; when fertilization is complete , the oocyte becomes a zygote . |
| NE($mix$)-RL | the expression fertilized ovum refers to a secondary oocyte that is impregnated by a sperm ; when fertilization is complete , the oocyte becomes a zygote . |
| STG | it is the beginning of a new human being . |
| Passage | sometimes the terms reactive airway disease and asthma are used interchangeably , but they are not the same thing . often , the term reactive airway disease is used when asthma is suspected , but not yet confirmed . reactive airway disease in children is a general term that does n't indicate a specific diagnosis . it may be used to describe a history of coughing , wheezing or shortness of breath triggered by infection . these signs and symptoms may or may not be caused by asthma . |
| Query | symptoms of reactive airway disease in adults |
| Ground-truth | coughing , wheezing or shortness of breath triggered by infection . |
| PLM | symptoms of reactive airway disease in children is a general term that does n't indicate a specific diagnosis . |
| NE($mix$)-MLE | symptoms of reactive airway disease in adults are a general term that does n't indicate a specific diagnosis . |
| NE($mix$)-RL | yes , reactive airway disease in children is a general term that does n't indicate a specific diagnosis . |
| STG | coughing , wheezing or shortness of breath triggered by infection . |
| Passage | sulfate is an anion , composed of a sulfur atom and four oxygen atoms . the overall charge of this anion is -2. the molecular formula for sulfate is so 4 2-. there , two oxygen atoms are bonded to the sulfur atom through double bonds and other two are bonded to sulfur through single bonds . the oxidation state of sulfur in sulfate is +6 . |
| Query | is sulfate sulfur and oxygen bonded |
| Ground-truth | yes , sulfate is sulfur and oxygen bonded . |
| PLM | no answer present . |
| NE($mix$)-MLE | yes , sulfate is sulfur and oxygen bonded to the sulfur atom through double bonds and other two are bonded to sulfur through single bonds . |
| NE($mix$)-RL | yes , sulfate sulfur and oxygen bonded to the sulfur atom through double bonds and other two are bonded to sulfur through single bonds . |
| STG | yes , sulfate is sulfur and oxygen bonded to each other through double bonds and other two are bonded to sulfur through single bonds . |

Table 12: Question Answering examples.

| Passage | gryffindor house is one of the four houses at hogwart into which the students are sorted when they start their first year . gryffindor is the hogwarts house of founder godric gryffindor . gryffindor house places high value on traits such as courage , chivalry , and nerve . during harry ' s time at hogwarts , the head of gryffindor house was minerva [ . . . ] |
|---|---|
| Query | what kind of house is a gryffindor |
| Ground-truth | gryffindor house is one of the four houses at hogwart into which the students are sorted when they start their first year . |
| PLM | gryffindor house is one of the four houses at hogwart into which the students are sorted when they start their first year . |
| NE($mix$)-MLE | no answer present . |
| NE($mix$)-RL | no answer present . |
| STG | gryffindor house is one of the four houses at hogwarts into which the students are sorted when they start their first year . |
| Passage | the median home value in louisville is $ 119,500 . louisville home values have gone up 1.8 % over the past year and zillow predicts they will rise 3.0 % within the next year . the median list price per square foot in louisville is $ 125 . the median price of homes currently listed in louisville is $ 180,000 . the median rent price in louisville is $ 1,050 . foreclosures will be a factor impacting home values in the next several years . in louisville 0.1 homes are foreclosed ( per 10,000 ) . |
| Query | average per therm price in louisville ky |
| Ground-truth | the median home value in louisville is $ 119,500 . |
| PLM | the average per therm price in louisville is $ 119,500 . |
| NE($mix$)-MLE | $ 125 per square foot |
| NE($mix$)-RL | the average per square foot in louisville is $ 125 . |
| STG | the median home value in louisville is $ 119,500 . |
| Passage | the brain is made up of more than just nerve cells ( neurons ) . although there are about 100 billion neurons in the brain , there may be about 10 to 50 times that many glial cells in the brain . neurons have two processes called axons and dendrites .... glial cells have only one . 2 neurons can generate action potentials ... glial cells can not . 3 however , glial cells do have a resting potential . 4 neurons have synapses that use neurotransmitters ... glial cells do not have chemical synapses . |
| Query | number of glial cells vs neurons |
| Ground-truth | 10 to 50 times . |
| PLM | no answer present . |
| NE($mix$)-MLE | $ 10 to 50 times that many glial cells in the brain . |
| NE($mix$)-RL | no answer present . |
| STG | 10 to 50 times |

Table 13: Text Summarization examples.

| Article | -lrb- cnn -rrb- – three runners died sunday during the detroit free press/flagstar marathon in detroit, michigan, police told cnn. an emt vehicle is at the scene sunday in detroit after three runners collapsed at a marathon. all three deaths occurred between 9 and 9:20 a.m. et, second deputy chief john roach said. a man in his 60s fell and hit his head, roach said. the cause of the fall was unknown. the man was transported to detroit receiving hospital, where he was pronounced dead. two other men, ages 36 and 26, also collapsed during the race and were pronounced dead at the hospital, roach said. all three collapsed near the end of the race, he said. witnesses describe scene " the weather at the time was overcast, roach said, with temperatures in the low 40s. [...] |
|---|---|
| Ground-truth | second deputy chief john roach : all three deaths occurred between 9 and 9:20 a.m. man in his 60s fell hit his head ; two men others , ages 36 and 26 , collapsed . race was detroit free press/flagstar marathon in detroit , michigan . |
| PLM | three runners collapsed at a marathon in detroit , police say . the cause of the fall is unknown . |
| Non-STG-MLE | three runners collapsed at a marathon sunday , police say . the cause of the fall is unknown , police say . |
| Non-STG-RL | three runners collapsed at a marathon sunday , police say . the cause of the fall is unknown , police say . |
| STG | three runners collapsed at a marathon in detroit , michigan . all three deaths occurred between 9 and 9:20 a.m. et . a man in his 60s fell and hit his head , police say . |
| Article | london, england -lrb- cnn -rrb- – up to 1,000 human rights campaigners demonstrated saturday in front of no. 10 downing street, the official residence of british prime minister gordon brown, calling on the british government to demand that full democracy be restored in pakistan. jemima khan, center, ex-wife for former pakistani cricket star imran khan, joins protesters in london. protesters waved placards and chanted in support of the resignation of pakistani president pervez musharraf, a week after he imposed a state of emergency in the country. the crowd of demonstrators massed behind barriers and included jemima khan, the ex-wife of former pakistani cricket star turned politician imran khan. the demonstrators carried placards saying " free the innocent " and " end musharraf's regime " and waved pakistani flags. imran khan, who heads the the movement for justice party, has been under house arrest since the emergency declaration. his ex-wife delivered a petition to a doorman at downing street, calling on britain to use its influence to ensure that all institutions are in place well in advance of pakistani elections originally scheduled for early next year. the petition also [...] |
| Ground-truth | human rights campaigners demonstrate in front of no . 10 downing street . protests urged uk government to demand full democracy restored in pakistan . cricketer turned politician imran khan ś ex wife jemima among protesters . |
| PLM | pakistani president pervez musharraf has been under house arrest since the emergency declaration . his ex-wife delivered a petition to a doorman at downing street . |
| Non-STG-MLE | pakistani president pervez musharraf has imposed a state of emergency in the country . he has been under house arrest since the emergency declaration . |
| Non-STG-RL | pakistani president pervez musharraf has imposed a state of emergency in the country . he has imposed a state of emergency in the country since last week . |
| STG | the official residence of british prime minister gordon brown . hundreds of protesters demonstrate in front of no . 10 downing street . the petition calls for full democracy in pakistan . |

Table 14: Text Summarization examples.

| Article | -lrb- cnn -rrb- spoiler alert! it's not just women getting cloned. that was the big twist at the end of " orphan black's " second season. the kickoff to the new season leads the list of six things to watch in the week ahead. 1. " orphan black, " 9 p.m. et, saturday, april 18, bbc america. the cloning cult sci-fi series remains one of the most critically acclaimed shows on tv, thanks in large part to the performance of tatiana maslany, who has taken on at least six roles on the show so far, including a newly introduced transgender clone. maslany told reporters this week that we can expect even more impressive scenes with multiple clones. [...] 2. " turn : washington's spies, " 9 p.m. et, monday, amc. the series about spies in the early days of the revolutionary war returns with a new subtitle, " washington's spies, " and a new monday night time slot. series star jamie bell told cnn what we can expect in the second season. " this year we have a lot more battles ; [...] 3. " game of thrones, " 9 p.m. et, sunday, hbo. the world of westeros returns for a fifth season in one of the biggest season premieres of the year. click here for more on what to expect. [...] |
|---|---|
| Ground-truth | critically acclaimed series " orphan black " returns . " turn : washington ś spies " starts a second season . " game of thrones " is back for season five . |
| PLM | series returns with a new subtitle , " washington ś spies , " and a new monday night time slot . series star jamie bell told cnn what we can expect in the second season . |
| Non-STG-MLE | " orphan black " returns with a new subtitle and a new monday night time slot . series star tatiana maslany is blown away by the response to the series so far . |
| Non-STG-RL | " orphan black " returns with a new subtitle and a new monday night time slot . series star jamie bell says the series will bring more recognition to the culper spy ring . |
| STG | the new season of " orphan black " premieres on saturday the 18th . " game of thrones " returns for a fifth season . |
| Article | energy drinks can pose a real problem for children, according to a new study from the american heart association. researchers found that 40 % of the 5,156 calls to poison centers for " energy drink exposure " involved children under age 6. in most of the cases, the parents didn't know the children had gotten hold of an energy drink. many of the calls reported the children were experiencing serious side effects, such as an abnormal heart rhythm, or they were having a seizure. the study is being presented at the american heart association's scientific sessions this week. study : caffeine affects teen boys more. study author dr. steven lipshultz has handled cases involving children who became sick after consuming energy drinks. while studies about the impact of caffeine on children are limited, lipshultz, [...] |
| Ground-truth | 40 % of " energy drink exposure " calls to poison control centers are for children under 6. study authors believe the number is under-counted . the fda does not set a caffeine limit for children . |
| PLM | drug calls to poison centers for " energy drink exposure " in most cases , the parents did n't know the children had gotten hold of an energy drink . many of the calls reported the children were experiencing serious side effects , such as an abnormal heart rhythm . |
| Non-STG-MLE | drug calls to poison centers for " energy drink exposure " in most cases , the parents did n't know the children had gotten hold of an energy drink . many of the calls reported the children were experiencing serious side effects , such as an abnormal heart rhythm . |
| Non-STG-RL | " this is a very concerning finding , " dr. laurence sperling says . the american academy of pediatrics recommends children consume no caffeine . |
| STG | drug calls to poison centers for " energy drink exposure " nearly 40 % of calls to poison centers for " energy drink exposure " involved children under age 6. study : caffeine affects teens more . |

