# OpenReview forum: "Selective Token Generation for Few-shot Language Modeling"
_ICLR.cc/2022/Conference — ICLR 2022 Submitted_

### Official Review · Reviewer_mErH · 2021-11-01

**Correctness:** 3
**Technical Novelty And Significance:** 2
**Empirical Novelty And Significance:** 2
**Recommendation:** 3
**Confidence:** 4

**Main Review:**

The reformulation of the proposed idea (selector to switch generation between a base PLM and a task-specific PLM) as a reinforcement learning problem is interesting. However, I have some major concerns on the technical clarify/rigor and experiment presentations that hopefully authors could clarify and help me better understand your contributions.

---

On the technical side:
- during optimization, the authors seem to be computing the reward at each time step (equation (9)) of the generation instead of on the whole-sequence level. However, as far as I know (I may be wrong), when RL methods are applied to NLG, the reward is typically computed on the whole-sequence instead of on the sub-sequence at each time of the generation. If the reward is computed on the partial generated sequence, how is it actually implemented? Why not compute the loss on the entire sequence which I think makes more sense for the BLEU and ROUGE metrics because both are operated on the whole-sequence level?
- What are the instantiations of $Q$, $A$, and $V$? From my understanding, $Q$ and $V$ are actually not computed and $A$ equals the metric score such as BLEU score.
- The selector module performs hard sampling, i.e., outputting either 1 or 0, in equations (5), (6), and (7). However, during _actual_ optimization, the selector performs soft sampling, i.e., outputting a probability between [0,1]. Why is there a mismatch? If the softmax is required as an approximation to the hard sampling, why do the authors need it? From my understanding, if the authors choose to use RL, then the loss function does not necessarily need to be differentiable, so approximating $\pi_s$ as softmax may not be necessary? Which implementation (hard or soft sampling) did the authors use in the experiments?
- What is the "penultimate representations" in the paper? Is it the last layer hidden states of all previously generated tokens, the last layer hidden state of the last generated token, or all layer hidden states of the last generated token?
---
On the experiment side:
- Overall I think more discussion and analyses are needed. Without these additional results, it is difficult to get insights into why and how the method works. The authors provide some discussions but I think it is insufficient; these claims should ideally be supported by empirical results. More below.
- My main concern is Table 2, which does not seem to make sense to me. If I understand correctly, for the ROUGE metric, the higher number the better, and it cannot be a negative number. In Table 2, PLM, which is a baseline, achieves far better results than any other methods under comparison by a huge margin, including the authors own proposed method. Moreover, one ROUGE result is negative. What happened?
- There is another baseline that I think the authors should compare but did not. How about setting $A$ in equation (9) to 1.0 and just run MLE using equation (9) as the loss? To me, $A$, which results from the authors' RL formulation of the optimization problem, is in practice simply a scaling factor to the usual MLE loss if they did not involve RL. This comparison would suggest how useful is RL training compared to MLE.
- There are different BLEU metrics relying on unigram, bi-gram, etc, and similarly for ROUGE metric. How does the different choices of these variants of the metrics as the reward function impact results?
- What does the selector do and when does it switch? Some more insights would be really useful. For example, the authors can provide some results on which word in the generation is generated by the frozen PLM $\pi_{\rm LM}$ and which word is generated by the fine-tuned policy $\pi_a$.

**Summary Of The Paper:**

The authors proposes a method to improve large pre-trained language model's ability in few-shot language generation. The main idea is to freeze the large pre-trained language model (PLM), fine-tune a copy of this PLM on the task at hand, and use a selector to switch between generating the next token from the freezes PLM or the fine-tuned PLM. Instead of the commonly used maximum likelihood estimation, the authors cast the optimization problem as a reinforcement learning (RL) problem using the task evaluation metric such as BLEU or ROUGE as the reward. The authors perform experiments on few-shot data-to-text and text summarization settings, comparing to a few baselines (fine-tuned PLM and a few variants of the proposed method).

**Summary Of The Review:**

Because of my concerns in the rigor and clarify in both the technical approach and the experiments, the paper needs further development and is not yet ready for publication at ICLR. Therefore, I unfortunately cannot champion acceptance for this paper at this time.

---

> ### Author Response · Authors · 2021-11-20
> **Author Response for mErH**
>
> We thank Reviewer **mErH** for all the comments. Please also refer to the common questions above for the answer to the remaining responses.
>
> - **Questions on the technical side**
>     - **<Q1>** Yes, you are right. The reward is computed after the whole sequence is generated. STG is trained to maximize the expected return as shown in the intermediate-term in equation 8. If we set the discount factor $\gamma = 1$, all the tokens will be assigned to the same return (e.g. BLEU).
>
>     - **<Q2>** $V$ is predicted by the critic network while $Q$ is approximated from a sampled (simulated) sequence and the corresponding sentence score by the current policy. $A$ can be computed by $Q-V$.
>
>     - **<Q3>** During training, for sequence generation, we first perform the hard sampling $i_t \in {0,1}$ from $\pi_s$ and then sample a token at $t$ according to the selected policy. Here, in order to use a single value head for both policies (selection policy and token generation policy), we can reformulate this overall sampling as a sampling from the hierarchical policy where $\pi_s$ can be considered a prior. Therefore, this hierarchical policy is then incorporated to the policy loss. Note that $A$ in the policy loss (9) is precomputed without gradient flow from the (hard) sampled sequences and the remaining term can use the probability of the hierarchical policy in (9).
>
>     - **<Q4>** The last hidden layer before the output logits of the PLM.
>
>
> - **Questions on the experiment side**
>     - **<Q1, 5>** We add experiments on QA task (Section 4.4), analysis on why STG generates better (Section 4.7), and a study of the selector (Appendix E) to the revised paper.
>
>     - **<Q3>**  We expected that the MLE version of STG would produce the same issue of Non-STG-MLE (i.e overfitting and exposure bias). It can easily be collapsed to select the task-specific policy only since the gradient flows the additional model only and, unlike STG, there is no chance to exploit diverse paths during training in the teacher forcing manner.  We discuss this issue in Appendix E.
>
>     - **<Q4>** We use the reward function as the same metric of their evaluation following previous works (e.g. Rouge-L for Summarization [1]). We agree that the reward shaping may be important in improving the reinforcement learner for text generation. It will be considered in future work.
>     - [1] `A deep reinforced model for abstractive summarization, 2017`

---

### Official Review · Reviewer_PchS · 2021-11-01

**Correctness:** 2
**Technical Novelty And Significance:** 3
**Empirical Novelty And Significance:** 2
**Recommendation:** 5
**Confidence:** 4

**Main Review:**

Strengths:
- The method is novel and brings knowledge of reinforcement learning to the challenges of NLP.
- Paper is overall well written and easy to follow

Weaknesses:
- Gains with the method are limited, especially in summarization. Breath of results is also limited as the authors consider only two tasks
- I would have appreciated seeing human evaluations. It is not obvious from the few samples in the appendix that the method leads to qualitatively superior generations.
- Beyond the numerical gains, it is not clear why the method leads to better generations. For instance, it is not clear how the token predictions at each step differ between the baseline and the introduced method?

Questions:
- The paper seems to be built on the hypothesis that fine-tuning PLMs in low data regimes leads to overfitting. Yet, it is not obvious from the numbers for PLM in line Table 1 and 2. Thoughts?
- Another core hypothesis seems to be the intuition that there are “task-relevant parts in sequence generation”. I couldn’t find any definition of what “task relevant parts” are. How does it differ from learning the format of the task?
- Why use the fine tuned model as an initialization for the model with adapters (non-stg). It seems related but I don’t understand this sentence: “this fine-tuning phase can be skipped when the advanced networks were used in the PLM such as External Encoders or Adapters which can cover the large domain shift”.
- Table 2: are the numbers for non-stg, ne and stg rouge points or % difference with the PLM baseline?
- Could you elaborate on the robustness of the method? For instance, what is the standard deviation between multiple runs?

Additional Feedback:
- Typo - Equation (1) - What is \theta_g? I think you mean \theta_a.


**Summary Of The Paper:**

The paper is interested in the problem of exposure bias of sequential text generation: tokens are drawn from the data during training, while during inference, the tokens are sampled from the model’s distribution.
Specifically, the authors consider this problem in the context of few-shot learning, where only a few labeled examples are available.
The authors introduce a method that combines task-specific adapters and reinforcement learning-based training: at each token prediction, the model chooses between the distribution of the original pre-trained LM and the distribution induced by the task-specific adapters on top of the pre-trained LM. The next token is sampled according to the selected distribution.
The authors observed gains in two tasks (data-to-text generation and summarization).


**Summary Of The Review:**

The introduced method is somehow novel, but the experiments have very limited breadth and analysis don't go beyond discussion.
Gains are also limited, and it's not clear looking at the current results that the solutions proposed really answer the problems exposed in the introduction (exposure bias, task-relevant parts, overfitting in low-data regime).

---

> ### Author Response · Authors · 2021-11-20
> **Author Response for PchS**
>
> We thank Reviewer **PchS** for all the comments. Please also refer to the common questions above for the answer to the remaining responses.
>
> - **<Q1>** Fine-tuning is conducted only for few-epochs. It is a well-known strategy called 'early stopping' to avoid overfitting.  The overall performance is improved in Additive Learning. However, Non-STG can not guarantee the improvement since the additional model can accelerate the overfitting as discussed in Section 4.7 of the revised paper.
>
> - **<Q2>** The task-specific parts may depend on the given task. For example, In Data-to-Text, the parts would be information of given structured data. In QA, the parts may be the first few tokens of an answer (see Table 4 and Appendix E).
>
> - **<Q3>** This fine-tuning phase accelerates the learning of the adapter. This is particularly when the adaptation requires to cover the large domain shift between the specific task data and PLM's original training data (usually cannot access). Severe performance degradation was observed for all the tasks when we skipped the fine-tuning.
>
> - **<Q5>** We experiment 3 times for each few-shot size of the summarization task as described in Section 4.5 of the revised paper. We produce the standard deviation in Table 3 and Appendix B of the revised paper.
>
> - **<Why the solutions proposed really answer the problems exposed?>** Appendix E shows that RL resolves the exposure bias and overfitting in low-data regime. It is hard to learn task-general part in few-shot setting. Especially task-general part can be easily forgotten by overfitting as shown in Section 4.7.  Section 4.7 and Appendix E supports that STG resolves challenges (large action space and credit assignment) of Non-STG-RL.
>
> - <**Additional feedback>** Thanks for the correction.

---

> > ### Comment · Reviewer_PchS · 2021-11-29
> > **thank you for answering my questions**
> >
> > Thank you for answering my question and revising the paper (especially section 4.7).
> > After reading the comments and answers from other reviewers, I have decided to retain my original score.

---

### Official Review · Reviewer_3Ee3 · 2021-11-02

**Correctness:** 4
**Technical Novelty And Significance:** 2
**Empirical Novelty And Significance:** 2
**Recommendation:** 5
**Confidence:** 3

**Main Review:**

Overall the paper is well written, barring some unnecessary complexity in the description of the technique (details below), and the paper does a good job iterating through details of the method and considering some alternatives. However I find the results unconvincing, primarily in their improvement over the base method. There are also some questions I have with the proposed method and how it is supposed to solve the overfitting issue like the paper claims. As summary, I would say the paper is clear and the proposal is not something I am familiar with in the literature, but the improvement seems marginal.

**Detailed Comments:**

* **<Introduction>** You describe large-scale training data as needing high quality annotations, but much of the larger models are trained on unsupervised text. I think this should be clarified.

* **<Section 2.2>** Can you explain the footnote?

* **<Section 2.2>** How exactly is the approach motivated by auxiliary training?

* **<Sections 2/3>** Equations 6 and 7 are clear and convey the key aspect of the proposed method well (absent the RL loss), however the majority of the other questions seem more dense and wordy than necessary, and are hard to follow at times (what is G in equation 1?). I think the sections introducing the method could benefit from more clear and concise mathematical formation (i.e. equation 9 can be re-written with less repetition).

* **<Section 3>** The motivation for STG is to prevent overfitting., I understand why wanting to rely on the pretrained logits mostly and only sometimes use the finetuned ones would help ameliorate overfitting, what I don't understand is exactly how this instantiation helps. Why does the RL not learn to always use the finetuned logits downstream, since surely that would help on the finetuning loss?

* **<Section 4>** Prompt engineering is brought up, how well does that compare to the other methods discussed in terms of finetune performance?

* **<Section 4>** Why not try NE(sum)?

* **<Section 5>** Why does the hierarchical policy reduce the action space?

* **<Table 5>** STI should be replaced with STG.

**Summary Of The Paper:**

This paper proposes STG, a method of adding a task-specific head to a pretrained language model, and finetuning it with RL along with a policy selector that chooses when to add this policy-specific logits to the pretrained logits for a given downstream task, targetting the low data regime.

**Summary Of The Review:**

Overall the paper is well written and I am not aware of this exact method being used in the literature, but the improvement strikes me as marginal in results with unclear motivation.

---

> ### Author Response · Authors · 2021-11-20
> **Author Response for 3Ee3**
>
> We thank Reviewer **3Ee3** for all the comments. Please also refer to the common questions above for the answer to the remaining responses.
>
> - **<Introduction>** Yes, task-specific “unsupervised" data can be used to train a task-specific model, however it also requires a large amount of training data in general. Here, we focus on the problem when we have limited training data. We clarify the sentence in the revised paper.
>
> - **<Section 2.2>** h indicates an encoding function, and the output of it is the representation. We remove this footnote in the revised paper.
>
> - **<Section 2.2>** The auxiliary training is designed for shifting the output distribution according to the target task efficiently avoiding the risks of rigidity and catastrophic forgetting. Although the auxiliary training is particularly designed for maximizing the likelihood of the target task output, it also can take an advantage for RL since the adapter logits are nearly zero before training is advanced. Namely, it lets the task-specific conditional distribution start learning from the distribution of PLM, not a uniform distribution.
>
> - **<Section 3>** In few-shot training, the explicit use of PLM logits can efficiently reduce the fine-tuning loss especially when the adapter is light since the adapter can focus only on the task-relevant part in generation. RL learns to do this naturally by stochastic policy sampling if the policy selector is initialized to perform uniform sampling. If STG is trained using MLE, yes it can be easily collapsed to select only a task-specific policy. This is because the gradient flows the additional model only and, unlike STG-RL, there is no chance to exploit diverse paths during training in the teacher forcing manner. You can find some learning curves for STG-MLE in Appendix E of the revised paper.
>
> - **<Section 4>** We did not compare the propt based and zero-shot based models for the following reasons: 1) For the prompt based model such as GPT-3, there is a limitation of context length.  For example, the Data-to-Text which has the smallest few-shot data of the tasks requires the context length of 2,545 on average. 2) For the zero-shot based moel such as GPT-2, there is an extremely large gap between PLM's distribution and the task's one. For example, the perplexity of the QA valid dataset is 3.08e+32 (calculated from GPT2-medium).
>
> - **<Section 4>** Normalized version of NE(sum) is NE(mix). Naive ensemble produces a mixture of probability distribution. We also evaluate another naive ensemble strategy NE(random) that randomly selects a token policy at each step between $\pi_{a}$ and $\pi_{LM}$, however it shows lower performances than the others.
>
> - **<Section 5>** STG surrogates the role of token selection to the PLM when the selector decides to select the PLM's policy. At this time, the task-specific policy does not need to seek their action space.
>
> - **<Table 5>** Thanks for the correction.

---

### Official Review · Reviewer_K8FA · 2021-11-03

**Correctness:** 2
**Technical Novelty And Significance:** 1
**Empirical Novelty And Significance:** 1
**Recommendation:** 3
**Confidence:** 5

**Main Review:**

Overall speaking, this paper is somehow not qualified enough. Below are the main points:
1. The algorithm is the RL-based method, the difference is that the author introduces a policy selection policy to decide the generation policy. This is straightforward and not new to the NLP generation community. The novelty and technical contribution are truly limited. If this is not the case, the results are also not surprising to see the improvements. Therefore, it is hard to see the shining points.
2. Even the method is not interesting, the experimental comparisons are also not enough. The authors do not give a comprehensive study of the results and the comparison with other papers or baselines. This is again, not qualified.
3. The selective policy is introduced in a self-learning way, which is automatically learned by the same hidden states of generation tokens. Therefore, a study of the policy selection policy is required. However, it is missed, also for other parts in the model.
The authors are highly encouraged to make a major modification of the paper.

**Summary Of The Paper:**

This paper proposes a selective token generation method for additive learning under pre-trained language models. Specifically, the authors introduce a selective module to decide the generation policy of tokens, either from the LM generation policy or the RL generation policy. The experiments are conducted on data-to-text task and text summarization task. Results show the improvements over the traditional baselines.

**Summary Of The Review:**

The paper does not give enough information to claim the importance of this paper, neither the method nor the results.

---

> ### Author Response · Authors · 2021-11-20
> **Author Response for K8FA**
>
> We thank Reviewer **K8FA** for all the comments. Please also refer to the common questions above for the answer to the remaining responses.
>
> - **<Response to 1 and 2 of the review>** As far as we know, the proposed method has not been used previously for text generation, especially in few-shot setting, and we compare it with the baselines most commonly used including several ablations. Please let us know the previous works which exploit a similar idea that adaptively selecting generation policy for few-shot text generation. Also, please let us know the other baselines we have missed and need to be compared.

---

> > ### Comment · Reviewer_K8FA · 2021-11-22
> > **Not convinced**
> >
> > Let me quickly put several related works that are missing, at least technical level, and the simple policy selection.
> >
> > [1] Bridging the Gap between Training and Inference for Neural Machine Translation. ACL 2019.
> >
> > [2] Incorporating Copying Mechanism in Sequence-to-Sequence Learning. ACL 2016.
> >
> > The RL-based methods are not novel in this paper, the selective way is straightforward as I mentioned.
> > I don't argue that it is the first time in specific few-shot text generation as the authors claimed, but a lot of relevant RL-based text generation methods should be incorporated.
> >
> > By the way, the performance gain is still limited, even for the updated Table 3. I do hope that the authors can give a full study/survey of the RL text generation fields.

---

> > > ### Author Response · Authors · 2021-11-23
> > > **Response to K8FA**
> > >
> > > Thanks for your comments.
> > >
> > > We would like to comment about the related works you mentioned first and then our thoughts on the novelty of the proposed method.
> > >
> > > - **<[1] Bridging the Gap between Training and Inference for Neural Machine Translation.>**
> > >     - From our understanding, a generator of [1] generates candidate sequences and it is assumed that among the already sampled candidates, there is a better sequence than the generated so far.
> > >
> > > - **<[2] Incorporating Copying Mechanism in Sequence-to-Sequence Learning.>**
> > >     - In few-shot setting, [2] has an issue when the copy pointer cannot produce the task-relevant parts since the PLM would produce task-general parts.  For example, there is no text *"no answer presented."* (highly likely that contains the task-relavant parts) in the given passage of the QA task.  Thus it makes the model performs only limited tasks such as Data-to-Text [3].
> > >     - [3] `Few-shot NLG with pre-trained language model, ACL 2020`
> > >
> > > - **<Novelty of the proposed method-1>** The proposed method should be distinguished from the studies mentioned above since it selects between two different generators (i.e. task-general and task-specific), neither already sampled sequences nor the input sequences. Moreover, the studies may hard to be the baselines for the following reasons:
> > >     - In few-shot tasks, the assumption of [1] does not fit since the generator can easily suffer from the problems (i.e. overfitting and exposure bias), and therefore the sequences would be collapsed or not help to guide the better generation (as discussed in the example of Section 4.7). (On the other side, [1] seems to be a higher-level concept than our works. It can be used over STG and will be considered in future work.)
> > >     - The copy mechanism [2, 3] is limited for the tasks. The STG is tested on the MLE objective similar to [2] and exposed to the problems as shown in Appendix E of the revised paper.
> > >
> > >
> > >
> > > - Here, let us quickly put some recent related studies of the RL text generation fields, especially in the perspective of exploitation of the PLM.
> > >
> > >     - [4] is a study of using an external guidance model that is built heuristically, such as topic classifier or non-preferred vocabulary, to guide text generation of the PLM with policy gradient.
> > >
> > >     - [5] is a type of Non-STG-RL we classified in this paper. They finetune the PLM with RL.
> > >
> > >     - [4] `Plug and play language models: A simple approach to controlled text generation. ICLR 2020`
> > >     - [5] `ReGen: Reinforcement Learning for Text and Knowledge Base Generation using Pretrained Language Model. EMNLP 2021`
> > >
> > > - **<Novelty of the proposed method-2>** The proposed method should be distinguished from [4, 5] and other studies (the RL text generation fields) referred to in this paper. This is because the proposed RL agents (i.e. the policy selector and the task-specific generator) interact with the environment which contains the task-general generator (i.e. PLM).  You can find a schematic illustration of the proposed method in Appendix D. Moreover, the above studies may hard to be the baselines for the following reasons:
> > >
> > >     - The external guidance model of [4] is hard to build for the tasks we have conducted. For example, we don't know how to build the guidance model that predicts the score (i.e. BLEU or ROUGE) at each time step during a generation.
> > >
> > >     - In few-shot setting, fine-tuning the PLM with RL as in [5] makes the PLM rapidly loses its capability due to catastrophic forgetting of the core knowledge of language (known as language drift) [6].
> > >
> > >     - [6] `Multi-agent Communication meets Natural Language: Synergies between Functional and Structural Language Learning, ACL 2020`
> > >
> > >
> > >
> > >
> > > We agree that the novelty may not be clear in the current version of the paper. We will clarify it and update the related works in the revised paper.

---

### Author Response · Authors · 2021-11-20
**Author Response for Common Questions**

We truly appreciate the reviewers’ insightful comments. We first address common concerns and then reply to each reviewer’s questions separately.

**Common Questions:**

- The improvements is marginal.
    - The improvement of the number may seem somehow marginal but it is noted that relatively simple neural adapter is used in this study (the study of the architectures will be conducted in future work). Hence, we claim that the performance “gain" of STG over the PLM needs to be compared with the other models. Roughly speaking, the Rouge-1 gain of STG is about 2 times larger than that obtained by Non-STG-RL (see Table 3 of the revised paper).
    - We have revised the paper including an empirical validation on the task of Question Answering (see Section 4.4, 4.6, and Table 2).  The STG significantly outperforms the Non-STG models as shown in the results.
- Why the method leads to better generations and how they works.
    - The revised paper contains more thorough analysis of the proposed STG in Section 4.7 and Appendix E, which includes generation examples, learning curves, performances without dynamic selection, and learned selection probability.
- Ambiguous numbers in Table 2 **[R-PchS, R-mErH]**
    - We apologize for the ambiguous denotation. The revised paper clarify the metrics (The Table 2 was moved to Table 3. see Table 3 and Appendix B).

---

### Author Response · Authors · 2021-11-21
**Summary of the second revision.**

Dear reviewers,

We uploaded our second revised version.

The changes compared to the first revision are as follows:
- **<Section 4.4, Table 2 and Appendix A>** We repeat the experiment of QA and produce the performance gain and standard deviation as in the experiment of Summarization.

---

### Decision · Program_Chairs · 2022-01-20

**Decision:**

Reject

**Comment:**

This paper presents a reinforcement learning inspired algorithm to train task-specific adapters to adapt pretrained language models for downstream tasks. The paper attempts to tackle an important problem. All reviewers have concerns about whether the results are strong enough to justify claims made in the paper. I appreciate revisions that have been done by the authors during the rebuttal period. However, I believe that the paper is still below the bar for ICLR. I recommend rejecting this paper.